# High Homogenization Pressures to Improve Food Quality, Functionality and Sustainability

**DOI:** 10.3390/molecules25143305

**Published:** 2020-07-21

**Authors:** José Mesa, Leidy Indira Hinestroza-Córdoba, Cristina Barrera, Lucía Seguí, Ester Betoret, Noelia Betoret

**Affiliations:** 1Institute of Food Engineering for Development, Universitat Politècnica de València, CP 46022 València, Spain; jomegu1@etsiamn.upv.es (J.M.); leihicor@doctor.upv.es (L.I.H.-C.); mcbarpu@tal.upv.es (C.B.); lusegil@upvnet.upv.es (L.S.); 2Grupo de Valoración y Aprovechamiento de la Biodiversidad, Universidad Tecnológica del Chocó. AA.292, Calle 22 No. 18B-10, Quibdó-Chocó CP 270001, Colombia; 3Instituto de Agroquímica y Tecnología de Alimentos, Consejo Superior de Investigaciones Científicas, 46980 Paterna, Spain

**Keywords:** high homogenization pressure, food functionality, bioactive components, agri-food waste, sustainability

## Abstract

Interest in high homogenization pressure technology has grown over the years. It is a green technology with low energy consumption that does not generate high CO_2_ emissions or polluting effluents. Its main food applications derive from its effect on particle size, causing a more homogeneous distribution of fluid elements (particles, globules, droplets, aggregates, etc.) and favoring the release of intracellular components, and from its effect on the structure and configuration of chemical components such as polyphenols and macromolecules such as carbohydrates (fibers) and proteins (also microorganisms and enzymes). The challenges of the 21st century are leading the processed food industry towards the creation of food of high nutritional quality and the use of waste to obtain ingredients with specific properties. For this purpose, soft and nonthermal technologies such as high pressure homogenization have huge potential. The objective of this work is to review how the need to combine safety, functionality and sustainability in the food industry has conditioned the application of high-pressure homogenization technology in the last decade.

## 1. Introduction

In the homogenization process, a fluid is forced to pass through a gap, causing energy transformations that directly affect the dissolved, dispersed or emulsified components. The fluid undergoes mechanical (shear, hydrodynamic and cavitation effects) stress and an increase in temperature (thermal effect) of approximately 2–3 °C for every 10 MPa of homogenization pressure [1]. These affect the fluid structure and properties, and also those of its constituent elements (particles, molecules, globules, droplets, aggregates, granules, etc.). Particle sizes decrease and more homogeneous distribution is achieved, facilitating operations such as mixing and emulsification. The effects are different to those induced by HPP (High Pressure Processing), in which prepacked food is loaded into a pressure vessel and then pressurized at a range of 100–1000 MPa, with water as the pressure-transmitting medium [2].

Initially, homogenization was introduced as a manufacturing step in the dairy industry. This operation reduced the size of fat globules, increasing the stability of the emulsion and, thus, the physical and chemical stability of milk. It had a great impact on the quality of dairy products such as condensed milk, curd or ice cream. The applied pressure was less than 30 MPa and it was applied in one or two steps. However, significant technological developments have occurred since then, having an impact on the design and geometry of homogenization valves, and making it possible to work at higher pressures and with very short processing times (a few seconds) [3]. High homogenization pressures were introduced at the beginning of the 2000s as an alternative, nonthermal treatment in the food industry, and applications were extended to industries other than dairy, e.g., to fields such as textile or biotechnologic.

The existence of valves of different geometries has given rise to the design of equipment that is able to work at pressures higher than 400 MPa. Thus, a distinction is made among standard homogenization for pressures between 0 and 50 MPa, high pressure homogenization (HPH) for pressures between 50 and 300 MPa and ultrahigh pressure homogenization (UHPH) for pressures equal to or greater than 400 MPa. Processing efficiency is modulated by applying various pressure ranges or combining a pressure value with a specific number of passes through the equipment [4]. In addition, the possibility of operating continuously for a great diversity of pumpable fluids has made it possible to extend applications to the activation/inactivation of enzymes, reduction of microbial load, mixing, dispersion, emulsification or encapsulation processes, cell breakage processes and the modification of proteins or macromolecules to obtain ingredients or additives with various properties.

Nowadays, concern about food functionality and sustainability is driving research interest in increasing the bioavailability and bioaccessibility of active components and probiotics, and in the extraction of macro- and micro- molecules from food byproducts. The challenges of increasing the nutritional characteristics of food must be combined with a reduction in environmental impact and increased food security. In this context, alternative, soft and nonthermal technologies such as high pressures homogenization have huge potential. The objective of this work is to review how the need to combine safety, functionality and sustainability has conditioned the application of high pressure homogenization technology in food. Advances and applications in the last decade have been organized according to the main challenges in the food industry.

## 2. Evolution and Major Applications in the Last Decade

Publications in peer-reviewed journals show that the main applications of HPH in food have the following objectives:Conservation and safety by decreasing the microbial load and inactivating enzymes. This occurs as a consequence of the thermal effect derived from mechanical stress or from structural changes in proteins.Recovery and extraction of proteins, fibrous materials and bioactive compounds (mainly polyphenols) and increase of the functionality considered in terms of technological use (stabilization of emulsions and dispersions, flow capacity and viscosity modifications, emulsifying activity improvement, etc.). Mechanical stresses and hydrodynamic effects induce cell disruption, favoring the release of intracellular content or structural components of the cell wall. Moreover, dispersed particles or fat droplets can be reduced in size and modified in structure.Increase of functionality in terms of health effect (increase bioaccessibility, bioavailability or probiotic effect). These effects result from favoring the release of bioactive compounds, the modification of biopolymer structures and the development of novel particle interactions and networking. Micro- or nano- capsules have also been developed.

In order to numerically quantify its evolution, the increase in the number of published scientific articles (in %) was calculated, taking into account the difference in the number of items between the last two decades. The results in each of the considered areas are included in Figure 1a,b. As shown in Figure 1a, between 2000 and 2009, HPH were used mainly for the extraction of proteins, although a large number of research works focused on microorganisms and enzymes inactivation, contributing to food preservation and safety. The last decade (from 2010 until now) revealed a significant increase (74.87%) in the total number of scientific articles published. The main areas in which there was an increase greater than the total value were the use of HPH for microorganism inactivation, fiber extraction, and above all, bioactive and probiotic components. The application of HPH to extract or increase the functionality of bioactive compounds, and to improve the probiotic effect, grew by 89% and 87.9% respectively (Figure 1a). The increasing interest among consumers and the food industry in improving the organoleptic and nutritional quality of foods, along with concern for the valorization of food waste, might explain this result.

Figure 1b shows the evolution in the number of published research works related to the application of HPH according to food type. Although the majority of works focused on fruit juices, the largest growth occurred in plant-based beverages and food waste. The huge increase in the consumption of plant-based beverages [5] and general concern about food waste-related issues are responsible for this increase. HPH technology is recognized as a green technology due to short processing times, low energy consumption, low CO_2_ emissions and the fact that it does not require polluting solvents.

This increase in research works based on HPH technology is also due to the development of new homogenization equipment that works at elevated pressures (i.e., up to 400 MPa) and supports specific conditions. Since the invention of adjustable valves in 1930 [6], the potential of homogenization technology has increased. The geometry and design of the valve determines the mechanical effect on the treated fluid. In 1982, the invention of the Gaulin Micro-Gap valve [7] greatly boosted the efficiency of the process, making high homogenization pressures possible in subsequent years and leading, more recently, to the development of ultrahigh homogenization pressure technology. In general, improvements have been obtained in all fields of application, making HPH an efficient tool with great potential for use in the food industry [8].

## 3. Preservation and Safety

HPH treatment for enzyme and microbial inactivation has been used in recent years as an alternative to thermal processes that, in most cases, cause undesirable effects such as nonenzymatic browning, cooked flavor or degradation of valuable components (see Table 1 and Table 2). It has been demonstrated that HPH treatment at processing pressures higher than 100 MPa contributes to microbial load reduction and enzyme inactivation. As previously stated, the heating that occurs in homogenization (an increase of about 2.5 °C per 10 MPa), together with structural modifications of cell walls, are the main phenomena responsible for the inactivation of microorganisms or enzymes. The final impact of HPH on microorganism viability or enzyme activity depends on several factors such as processing pressure, microbial strain or enzyme and food matrix.

In general, it has been verified that gram-negative bacteria exhibit greater susceptibility to inactivation by HPH than gram-positive ones, due to the reduced content of peptidoglycan in the cell wall that makes it thinner, and therefore, easier to disrupt. Fungi and yeasts seem to have a susceptibility that is intermediate between gram-positive and gram-negative bacteria, probably due to their wall structure, which is thicker but more complex than that of gram-positive bacteria [9]. For the inactivation of bacterial spores, pressures up to 400 MPa and additional steps are required [8].

Significant work has been undertaken on the use of HPH to reduce microbial load in fruit juices. In this type of food, it has been observed that the presence of some aroma compounds and essential oils can greatly influence the effect of HPH treatment. Patrignani et al. [12] studied the effect of the number of passes and citral addition on the spoilage microbiota of apricot juice when subjected to HPH at 100 MPa. Their results showed that yeast cell viability decreased with the increase of passes, and the relationship between both variables followed a linear trend. Moreover, the citral addition enhanced the effect of HPH, increasing the storage time by 6–8 days. To analyze the effect of the food matrix, the same authors compared the effect of HPH treatment at 100 MPa on the viability loss of *S. cerevisiae* 635 inoculated at a level of about 6.0 Log10 cfu/mL in apricot juice and carrot juice. In apricot juice, a significant decrease in the viability of 2.2 logarithmic cycles per mL was obtained with only four repeated passes at 100 MPa. A further increase of the number of passes at 100 MPa did not significantly increase the effectiveness of HPH treatment. Concerning carrot juice, eight repeated passes at 100 MPa were unable to completely inactivate the inoculated cells. They concluded that because of the higher viscosity and sugar content, apricot juice required more passes in HPH treatment to reduce yeast load [26]. In contrast, *Zygosaccharomyces bailii* 45 exhibited the same susceptibility to HPH treatment in both juices. Eight passes at 100 MPa resulted in a yeast inactivation higher than 2.5 log CFU/mL, regardless of the juice considered [27]. Nevertheless, Benjamin and Gamrasni [16] showed that HPH treatment at 100 and 150 MPa was not sufficient to reduce total bacteria and yeast count in pomegranate juice; rather, it needed to be combined with a thermal treatment at 65 °C for 15 s to achieve the same effect as pasteurization at 75 °C.

Besides fruit juices, plant-based beverages are complex dispersions with suspended proteins and oil droplets that require a homogenization stage to stabilize them and extend their commercial life. HPH can be applied at pressures higher than 100 MPa using multiple passes for these purposes, along with microbial cells destruction [21,22]. Valencia-Flores et al. [20] compared the effect on bacterial growth of HPH at 200–300 MPa and soft temperature inlet (55–75 °C) with conventional pasteurization treatment (90 °C, 90 s) in an almond beverage. They showed that 200 MPa and an inlet temperature of 55 °C yielded better results than conventional pasteurization on microbiological quality.

Beer is another beverage which may be treated with HPH. Research has established that it is possible to completely inactivate microorganisms in addition to improving the color of beer by HPH at pressures between 200–300 MPa and with 1 to 3 passes. The addition of antimicrobials such as lysozyme enzyme (50 mg/L) had a synergistic effect, reducing the required pressure to 100–150 MPa. However, HPH treatment could result in greater values for turbidity, and it would be necessary to perform another stabilization treatments to minimize the negative effects [23,24].

HPH treatment has also been used to modulate the activity of various enzymes. This treatment can increase or decrease enzyme activity depending on the processing conditions (pressure and number of passes), homogenizing valve structure, specific enzyme, pH, temperature and food matrix. Since enzymes are a complex type of globular protein, the mechanical forces and cavitation effects associated with HPH treatment result in conformational and structural changes which modify enzyme activity and stability. The main modifications in enzymes are linked to changes in the quaternary, tertiary and even secondary structures. The formation or interruption of hydrogen bonds, Van der Waals, hydrophobic and electrostatic interactions can occur, increasing the number of hydrophobic sites, revealing amino acid and sulfhydryl groups, and thus, accelerating, delaying or impeding enzyme–substrate interactions [24]. Furthermore, the magnitude of the changes induced by the HPH treatment will determine their reversibility or irreversibility. Aguilar et al. [28] noted that protein denaturation can be reversible at 100 MPa, but that it is irreversible above 200 MPa.

In the case of juices, the main alteration reactions are caused by the polyphenoloxidase, which is responsible for browning and oxidation reactions; it was shown that it was possible to inactivate it with homogenization pressures of 80–150 MPa [29,30]. On the other hand, α-amylase—whose use in recent years has been increasing, since it reduces the starch content in beverages, thereby avoiding turbidity and gelatinization—is resistant to HPH [31]. A similar resistance was observed on *Pseudomonas fluorescens* protease when HPH at 100–150 MPa was applied to reduce its proteolytic rate [32].

On the other hand, HPH can also be applied to enhance enzyme activity. Some authors have applied HPH to increase the activity of enzymes involved in the shelf life or processing of several food matrices. Lysozyme and lactoferrin in milk increased their antimicrobial activity against *L. monocytogenes* after HPH at 100 MPa [8,36]. Pinho et al. [17] observed an increase in the enzymatic activity of lactoperoxidase in skim milk at pressures of between 100 and 250 MPa. In contrast, if the homogenization pressure increased up to 300 MPa, a reduction of around 30% in enzyme activity was detected. In another work, defatted peanut flour was dispersed in distilled water and pH adjusted, and further subjected to HPH treatment at 40 and 80 MPa. After that, the peanut protein was recovered from the dispersed solution by an acid precipitation and redispersed in distilled water. The HPH treatment increased the extraction yield and the hydrolysis of the peanut protein isolates by endogenous enzymes. DPPH radical scavenging and hydroxyl radical scavenging activities were also increased [34].

It was shown that low cost, versatility and performance improvement of enzymatic processes can be achieved when the activity of commercial enzymes is increased by HPH. In particular, Tribst et al. [24] improved the activity of amyloglucosidase, glucose oxidase and neutral protease using HPH between 100–150 MPa and nonoptimum temperatures. Commercial enzymes derived from fungi and available as powders were diluted in acetate buffer solutions and then subjected to HPH treatment. Tribst et al. [24] observed an uneven effect of the number of passes. Only one pass was required to increase the activity of amyloglucosidase and neutral protease, and while no effect was observed in subsequent passes, successive steps continued to increase the enzyme activity of glucose oxidase; the energy involved in the molecular changes associated with the increase in enzyme activity might be responsible for this.

## 4. Extraction and Technological Functionality Improvement of Proteins, Fibrous Materials and Bioactive Compounds

HPH has been used in recent years to contribute to food process sustainability [5]. In this area, HPH has been applied for the valorization of agrifood byproducts with two objectives: (i) to increase the extractability of intracellular or cell wall structural components, and (ii) to improve the technological functionality of biomolecules from food byproducts. Most agri-food wastes or byproducts are rich in fibrous material and, in some cases, in proteins or bioactive compounds which are of interest to the food industry for use as food ingredients [37] or as sustainable packaging materials [38]. HPH induces cells disruption, favoring the release of structural and intracellular contents.

The main kind of products with which HPH is used to extract fibers, proteins or bioactive compounds are solid byproducts, such as pomace from fruit or vegetable juicing, fruit or vegetable peels, minimal processing waste and vegetal parts of plants or cereal seed hulls; the most important examples are included in Table 3. In these cases, solid wastes need to be fluidized by diluting them in water or another solvent. In other cases, an extraction method is applied and the extracted phase is further subjected to HPH. The authors of [39] extracted pectin form potato peel by HPH at 200 MPa. They obtained improvements in the viscosity, emulsifying properties, degree of esterification and physicochemical characteristics, and therefore, recommended the use of HPH to obtain pectin that could be used as a stabilizing agent or a thickener in food manufacturing. Similarly, Fayaz et al. [40] showed that HPH favors the release of okara proteins and soluble fiber. Soy okara was dispersed in deionized water at 10 g/100 g and prehomogenized with a high-speed blender. After that, a homogenization pressure of 150 MPa for 5 passes made it possible to extract proteins with a yield of 90%. The authors of [41] applied HPH to make edible and biodegradable films for food packaging from a type of edible fungus, i.e., *Flammulina velutipes*. Wu et al. [42] demonstrated the possibility of using HPH treatment to make biodegradable biopolymer films from pomelo peel.

HPH reduces the particle size and structure of macromolecules, modifying their solubility, interaction properties, viscosity, or other physic-chemical properties. Saricaoglu et al. [43] improved the functionality of proteins from the hazelnut industry by HPH at 100 MPa and 1 pass. The homogenization pressure decreased the particle size of proteins, increasing their zeta potential and water solubility; emulsifying and sparkling properties were improved too. Hua et al. [44] demonstrated a microstructural change of tomato waste fibers by applying HPH at 100 MPa and 10 passes. The authors transformed around 8% of the insoluble fibers into soluble ones. Xu et al. [45] indicated that for the preparation of soluble peach fiber from fresh peach marc, it must be dispersed in three times the volume of deionized water, thus improving the efficiency of cellulose hydrolysis. For pectin extraction from milled dried lemon peel, variations in dilutions changed the properties of the extracted pectin, resulting in residues with different pectic characteristics [46]. Discarded external lettuce leaves were dispersed in hydroalcoholic solutions and polyphenols extracted with ethanol to obtain good phenolic extraction yields [47].

## 5. Increase of Bioavailability and Encapsulation of Bioactive Compounds

In the last decade, many studies have been carried out to demonstrate that the application of HPH to liquid foods can modify the bioaccessibility (i.e., the fraction of an ingested nutrient that is released from the food matrix and made available for intestinal absorption) or bioavailability (i.e., the fraction of an ingested nutrient that is absorbed by the intestine and incorporated into the bloodstream) of its bioactive compounds. In most studies, an increase in the bioaccessibility of phytochemicals was observed due to their release within the structure of the food in which they were found. In other cases, a modification of biological functionality occurs due to a change in its chemical structure. Zhou [62] carried out an interesting review that demonstrated these effects in three bioactive components: carotenoids, phenolic compounds and vitamin C. The review showed that fruit juices (carrot, tomato, orange, apple and berries) are the most common food in which HPH increases the bioaccessibility of bioactive compounds. HPH decreases the particle size of suspended pulp and increases cloud stability and, thus, the availability of bioactive components. Treatment with HPH in mandarin juices increased the bioaccessibility of total carotenoids by five times, although in the case of flavonoids, no such drastic changes were observed. Therefore, HPH treatment was recommended for the production of tangerine juices that promote health, mainly through the improvement of the bioaccessibility of the carotenoids contained therein [63]. Quan et al. [64] established that the improvement of bioaccessibility could be conditioned by the food matrix. They observed that HPH at 250 MPa favored the release from cell walls and increased the content of total phenolic compounds in kiwi and pomelo juices, but that it had a negative effect on its bioaccessibility (in vitro) as a consequence of the major degradation that occurred in the digestion process. Conversely, the addition of skimmed or whole milk to the juices had no significant effect on total phenolic content, but increased their bioaccessibility in kiwi juice and pomelo juice from 21.6% to 37.8% and 60.1% to 63.3%, respectively. Similarly, Alongi et al. [65] showed (in vitro) that the bioaccessibility of chlorogenic acids increased from nearly 25% to more than 50% by adding milk with different fat contents to coffee and applying HPH (50–150 MPa). Alongi et al. [65] observed that the pressure required was lower the lower the fat percentage; they attributed this effect to the micellarization of chlorogenic acids, a phenomenon that reduced their susceptibility to degradation during digestion. Sometimes, positive effects of HPH were observed after storage. Betoret et al. [66] found in low pulp mandarin juice that, despite the increase in suspended pulp after HPH and trehalose addition, flavonoid hesperidin initially decreased but resulted in less flavonoid degradation during storage.

HPHs have also been applied, albeit with a much smaller number of published articles, to nondairy, vegetable-based beverages. Although, in some cases, no significant improvements in the nutritional characteristics were detected, in others, such as almond or soy beverages, a reduction in antinutrients was achieved [67]. The denaturation, aggregation and chemical modification of proteins may change their allergic potential. Toro- Funes et al. [68] demonstrated an increase of 40% in the extractability of phytosterols from almond milk subjected to a HPH of 300 MPa (6 passes). However, the content of tocopherol and polyamines such as spermidine were reduced by up to 90%. The application of HPH for kefir production from a hazelnut beverage achieved improvements in the total content of phenolic compounds and antioxidant capacity, causing a reduction in the content of lactic and citric acid [69].

Improvements in the bioavailability of bioactive components by HPH are also possible in solid foods. HPH (10–20 cycles and 100–200 MPa) was used to fabricate an aqueous nanosuspension of fermented soybean powder, favoring in vitro release of isoflavones from nanosuspension [70].

The other way in which HPH can contribute to improvements in the nutritional properties of foods is by using this technology for the encapsulation of bioactive components (see Table 4). In this way, it is possible to increase the stability, conservation and controlled delivery to target sites, thereby increasing food functionality. HPH produces intense disruptive forces that break up particles into smaller sizes, favoring the encapsulation of specific components in a suitable media. Mechanical stress and heating associated and emulsifier interactions can affect the effectiveness of the process, and therefore, the activity and bioaccessibility of the bioactive compound.

Many studies have investigated the nanoencapsulation of curcumin by HPH at 40–100 MPa. The results showed that the emulsifier type had an influence on the bioaccessibility of curcumin [71]. Frank et al. [72] studied the degradation of anthocyanins from bilberry extract by subjecting them, by HPH, to temperature and mechanical stresses similar to those involved in the process of emulsification and encapsulation. HPH was applied with a simple pass and in a pressure range between 30 and 150 MPa. Thereafter, the samples were immediately cooled to 298 K. The results showed no significant influence of mechanical stresses associated with HPH on anthocyanin stability, even at high-pressure treatment, i.e., up to 150 MPa. The combination temperature–time was the main parameter affecting the degradation of anthocyanin.

A great level of interest has been shown in applying HPH as an encapsulation technique in bacteria with probiotic effects. Patrignani et al. [73] underlined the potential of the HPH microencapsulation of probiotic microorganisms to produce fermented milk with improved functionality and enhanced sensory properties. They established 50 MPa and 5 passes as being adequate conditions to produce stable microcapsules of Lactobacilli with high yield and excellent viability during storage. Moreover, the microencapsulation of adjunct bacteria reduced the acidity of fermented milk. Calabuig-Jiménez et al. [74] microencapsulated *L. salivarius* spp. *salivarius* in alginate coatings using HPH at 70 MPa. A positive effect of microcapsules was observed when evaluating the survival of the probiotic strain on simulated gastrointestinal conditions.

HPH treatments at pressure levels below 100 MPa, considered sublethal pressures, were applied to microbial cultures, i.e., initiators, co-initiators, or probiotics and yeasts, in order to produce cultures with improved functional, technological and sensory properties (some examples are included in Table 5). The use of strains belonging to the genus *Bifidobacterium* and *Lactobacillus* predominated as probiotics, and to a lesser extent Enterococcus, Streptococcus and Saccharomyces [8]. The bacterial cells responded to mechanical stress induced by HPH, modifying their metabolic activity and membrane composition. Therefore, technological and functional properties such as fermentation kinetic, enzymatic activities, hydrophobicity or resistance to gastrointestinal digestion can be improved.

Siroli et al. [83] reported that the main regulatory mechanism that probiotic lactobacilli adopt to counteract pressure stress is the modification of the composition of membrane fatty acids. Specifically, they observed an increase in unsaturated fatty acids when HPH at 100 and 150 MPa was applied to *Lactobacillus paracasei* A13. Considering that the increase of the unsaturation level is a key mechanism to compensate for the oxidative damages induced by physico-chemical stressors in microbial cells, they concluded that HPH at sublethal pressures is useful to improve the activity of some Lactobacillus species.

Lanciotti et al. [84] studied the effect of HPH between 50 MPa and 100 MPa on the fermentation kinetics, metabolic profile and enzymatic activity of four species of Lactobacilli involved in dairy product fermentation and ripening. Although the results varied according to the species, they documented no significant effect on cell viability, an increased proteolytic activity and positive changes in fermentation dynamics. The resistance to simulated gastric conditions, hydrophobicity and auto-aggregation capacity were strain-dependent for *L. acidophilus* Dru and *L. paracasei* A13 when subjected to HPH at 50 MPa. HPH increased the three properties for *L. paracasei* A13 but reduced them for *L. acidophilus* Dru; the authors attributed this to the compositional and structural differences in the cellular outer structures, thus suggesting that the HPH effects on macromolecules and their interactions with the gut immune cells play a key role in the probiotic effect. The same authors noted that HPH treated *L. paracasei* cells modified their interaction with the small intestine of mice, inducing a higher IgA response compared to untreated *L. paracasei* cells [12,79]. Betoret et al. [66] demonstrated an improvement in the hydrophobicity of *Lactobacillus salivarius* spp. salivarius added to mandarin juice with trehalose when HPH at 0, 20 and 100 MPa was applied.

## 6. Conclusions

Although in the beginning, the application of high homogenizing pressures was aimed at more efficient homogenizing and increasing the stability of emulsions such as milk, advances in valve design have allowed for an increase in working pressure extending the scope of possible applications.

In the last decade, the number of research works related to the implementation of HPH—for extracting bioactive components from agri-food wastes, to improve the bioavailability and probiotic properties of bioactive components and microorganisms, and also as an encapsulation technique—has grown by more than 80%. At the same time, progress has been made in the application of HPH to reduce the microbial load or modulate the activity of some enzymes.

The general mechanisms responsible for the effect of HPH are known, but the final effect is largely conditioned by the type of valve, pressure applied, number of passes, the nature of the components and macromolecules, and the food matrix. For this reason, research is needed for each specific application.

Results published in the last decade have shown HPH to be a nonthermal technology which is able to accomplish the food industry’s objectives of quality and safety, functionality and sustainability.

## Figures and Tables

**Figure 1 molecules-25-03305-f001:**
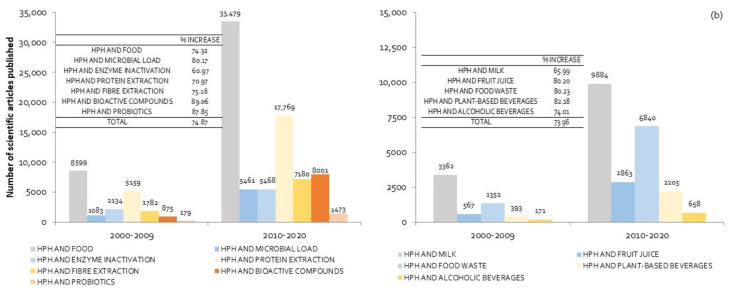
Number of scientific articles published according to areas of application of HPH in the food industry (**a**) and the different types of food with which it has been used (**b**). % INCREASE was calculated as the difference in the number of articles published between 2010–2020 and 2000–2009, divided by the number of articles published in the period 2000–2009, expressed as a percentage. (Source: Science Direct. The following keywords and their combinations were used as the main search terms: high homogenization pressures, nonthermal technologies, food processing, encapsulation, functional food, bioactive components, probiotics, microbial load, enzyme inactivation, protein extraction, milk, fruit juice, food waste, plant-based beverages and alcoholic beverages).

**Table 1 molecules-25-03305-t001:** Research works evaluating the decrease in microbial load in different food products by HPH.

Product	Treatment	Terms	Microbiologic Control	Results	Reference
Fruit juices (apricot and carrot)	HPH + rapid cooling	100 MPa (1–8 passes)	*Zygosaccharomyces bailii 45*	The juice type affected the yeast fate (growth or death) and viscosity change after HPH treatment.	[8]
Mango nectar	HPH + thermal shock	200 MPa 10–20 s at 60–85 °C	*A. niger* (COI 4573)	The combination of HPH with subsequent thermal shock was efficient in inactivating heat resistant mold in mango nectar.	[10]
Banana juice	HPH + rapid cooling	0, 150, 200, 300 and 400 MPa	Total mesophilic bacteria	Pressures greater than 200 MPa were required to obtain a reduction of four logarithmic units.	[11]
Apricot juice	HPH + citral + rapid cooling	100 MPa (1,3,5 and 8 passes)	*Saccharomyces cerevisiae* SPA	Decrease of the viability of the yeasts following a linear tendency with pressure. Improvement of the antimicrobial effect by adding citral.	[12]
Mango juice (*Mangifera indica L*.)	HPH + heat treatment	40–190 MPa (1–5 passes)	Total plate count, molds and yeasts	Complete inactivation of molds and yeasts was achieved by one and three passes at 190 MPa and 60 °C, while the total plate count was less than 2.0 log CFU/mL.	[13]
Mulberry juice (*Morus atropurpurea* Roxb.)	HPH + heat treatment + Addition of Dimethyl Dicarbonate (DMDC)	200 MPa (1–3 passes)	Total count, yeast, mold and lactic acid bacteria	Combination treatment with three passes at 200 MPa and 250 mg DMDC/L decreased total count to the level reached by heat treatment at 95 °C.	[14]
Lupine based drinks	HPH + refrigeration	50, 100 and 175 MPa (2,4,6 passes)	Total bacterial count, molds and yeasts. *Bacillus cereuses* and coliform bacteria	At 175 MPa, yeasts, molds and coliforms were completely eliminated with two and four passes	[15]
Granada juice	HPH + low temperature pasteurization	100, 150 MPa (10 passes) 55 or 65 °C for 15 s	*Escherichia coli* (ATCC 26) and *Saccharomyces pastorianus* (ATCC 42376)	HPH at 150 MPa followed by a low heat intensity at 65 °C for 15 s showed a reduction of 3 log CFU/ mL.	[16]
Skim milk	Heat treatment + HPH	100–300 MPa	*Bacillus stearothermophilus* ATCC 7953 and *Clostridium sporogenes* PA 3679	The efficacy of HPH is similar to pasteurization and must be combined with other conservation techniques.	[17]
Milk	Heat treatment + HPH	300 MPa	Spores of *B. cereus*, *B. lincheniformis,* *B. sporothermodurans,* *B. coagulans,* *B. stearothermophilus,* *and B. subtilis*	Sterility at 300 MPa can be achieved with an initial milk temperature of 85 °C.	[18]
Skim and whole milk concentrates	Heat treatment + HPH	Skim milk: 0,20,50,70, 100,120 and 150 MPa. Whole milk: 0,20,30,35 and 40 MPa.	Total count, coliforms, enterobacteriaceae, molds and yeasts and *Staphylococco*	HPH at 120 MPa completely inactivates the microbial load of milk concentrates.	[19]
Almond beverages	Heat treatment + HPH	200, 300 MPa (1,2 passes)	*Micrococcaceae, Bacillus cereus* and Mesophilic aerobic bacteria	Complete elimination of microbial growth when working with the highest pressure and with an inlet temperature of 65–75 °C.	[20]
Rice drink	HPH+ sonication	50–100 MPa (1–3 passes)	*Lactobacillus Plantarum, Lactobacillus Casei, y Bifidobacterium Animalis*	Reduction and elimination of postacidification by lactic acid bacteria.	[21]
Tiger nuts’ milk beverage	HPH + refrigeration	200 and 300 MPa	Psychotropic bacteria, Lactobacilli, Enterobacteriaceae and fecal coliforms	Improved shelf life and microbial inactivation compared to other heat treatments.	[22]
Lager beer	HPH + lysozyme addition	0–300 MPa	*Lactobacillus brevis* (CCT 3745)	The inhibitory concentration of lysozyme against *L. brevis* was 100 mg/ L. HPH at 100, 140 and 150 MPa promoted decimal reductions of 1, 3, and 6 in microbial counts.	[23]
Pilsen beer	Heat treatment + HPH	100, 150, 200 and 250 MPa (1–3 passes)	*Lactobacillus del brueckii*	It is possible to inactivate the most common microorganisms that cause beer deterioration at 250 MPa. The effect increases with increasing the number of passes.	[24]
Wine	Chemical treatment + HPH	0, 50, 100 and 150 MPa	*Saccharomyces bayanus*	HPH at 150 MPa was the best treatment, inducing yeast autolysis; also suitable for the acceleration of sur lie maturation.	[25]

**Table 2 molecules-25-03305-t002:** Research works evaluating enzyme activity modulation by HPH.

Product	Enzymes	Treatment	Effect	Reference
Commercial enzymes	Glucose oxidase	50, 100, 150 MPa	Decrease in enzyme activity at 50 MPa. Improvement in activity and stability at 100 and 150 MPa	[33]
Commercial enzymes	Amyloglucosidase, Glucose oxidase, Neutral protease	Amyloglucosidase, neutral protease: 150, 200 MPa (3 passes). Glucose oxidase: 100, 150 MPa (3 passes)	Improvement of enzymatic activity	[24]
Fruit juices	α-amilase	0, 40, 80, 120 and 150 MPa	Stability of the enzyme	[31]
Apple juice	Polyphenoloxidase	150 MPa (10 passes)	Inactivation	[29]
Lettuce waste juice	Polyphenoloxidase	80 MPa (1 pass) and 150 MPa (1–10 passes)	Inactivation	[30]
Peanut protein	Alcalase	0, 1, 40 and 80 MPa	Increased enzymatic hydrolysis.	[34]
Chicken egg white	Lysozyme muramidase	40, 80, 120, 160 and 190 MPa	Activation and increase of enzymatic activity.	[35]
Raw skim milk	Alkaline phosphatase and lactoperoxidase	100, 150, 200, 250 and 300 MPa	Decrease and inactivation of alkaline phosphatase. Increased activity of lactoperoxidase.	[17]
Milk	Protease *Pseudomonas fluorescens*	100 and 150 MPa	Decreased proteolytic rate	[32]

**Table 3 molecules-25-03305-t003:** Research works aimed to the extraction and improvement of technological functionality of proteins, fibers or bioactive compounds from agri-food wastes by HPH.

Substrate	Component	Treatment	Objective	Reference
Sweet potato leaves	Flavonoids	100 MPa (2 passes)	Strengthens the antioxidant activities of the flavonoid.	[48]
Potato peel	Biopolymer film	150 MPa	Extraction	[49]
Peach pomace	Soluble fibers	140 MPa (4 passes)	Significantly improved the efficiency of cellulase hydrolysis in the preparation of soluble fibers and a high binding capacity for sodium cholate and cholesterol.	[45]
Potato peel	Phenolic acids	159 MPa (2 passes) + NaOH treatment	Improved extraction and release of total phenolic content and total flavonoid content.	[37]
*Desmodesmus* sp. F51	Carotenoids	69–276 MPa (1–4 passes)	Extraction	[50]
Dry tomato residue waste	Fibers	100 MPa (10 passes)	Improved the soluble fiber content and its oil holding capacity.	[44]
Citrus peel	Fibers	90, 160 MPa (2 passes)	Improvement of physical, chemical and functional properties including surface area, water holding capacity, texture and viscosity.	[51]
Lemon peels fiber	Pectin	20 and 80 MPa	Extraction	[46]
Soybean	Protein	100 MPa	Extraction	[52]
Hazelnut oil industry by-products	Hazelnut meal proteins	0, 25, 50, 75, 100 and 150 MPa	Improves functional (solubility, emulsifying and foaming properties) and rheological properties of proteins.	[53]
Black cherry tomato waste	Pectin	0, 40, 80, 120 and 160 MPa (2 passes)	Increase the esterification degree of pectins.	[54]
Carrot processing waste	Biodegradable composite films were prepared	138 MPa (7 passes)	Extraction	[55]
Lettuce waste	Polyphenols	50, 100 MPa	Extraction	[47]
Potato peel	Pectin	200 MPa	Increased galacturonic acid content, viscosity and emulsifying properties. Decreased esterification degree and molecular weight.	[39]
Broccoli seeds	Sulforaphane	20–160 MPa (1–5 passes)	Increases the extraction yield.	[56]
Agri-food waste (tomato peel, coffee beans)	Application for structuring peanut oil	70 MPa (3 passes)	Replacing part of the lipids with water and low calorie fibers.	[57]
Edible mushroom by-products	Biodegradable edible film	100 MPa (3 passes)	Improve tensile strength, elongation at break, water vapor permeability, oxygen barrier and thermal stability.	[41]
Grape seeds, tomato stem, walnut shells, coffee	Polyphenolic compounds and antioxidants	20, 50, 100, 120 MPa	Extraction	[58]
Soybean okara	Proteins and soluble fibers	50, 100, 150 MPa (1 pass) 150 MPa (5 pases)	Extraction	[40]
Sugar palm	nanofibrillated cellulose	50 MPa (3 passes)	Extraction	[59]
Tomato peels	Bioactive compounds: proteins, polyphenols, lycopene	100 MPa (1–10 passes)	Increased release of intracellular compounds (proteins, sugars, antioxidants)	[60]
Pomelo peel	Biopolymer film	20, 40, 60 and 80 MPa (10 passes)	Improved mechanical properties, microstructure, optical and barrier properties.	[42]
Soybean meal	Resins	20 MPa	Extraction	[61]

**Table 4 molecules-25-03305-t004:** Research works in which HPH treatment was applied to encapsulate.

Component Encapsulated	Matrix	Conditions	Results	Reference
*Lactobacillus paracasei* A13 and *Lactobacillus salivarius* subsp. *salivarius* CET 4063	Fermented milk	50 MPa (5 passes)	The microcapsules presented high yields in terms of trapped viable cells and acceptable sizes. Furthermore, microencapsulation caused a decrease in acidity in fermented milk.	[73]
Phenolic compounds and anthocyanins from blueberry pomace	-	50–200 MPa	The encapsulation efficiency, size and charge characteristics of the emulsion droplets were affected by HPH.	[75]
*Lactobacillus salivarius* spp. *Salivarius*	Mandarin Juice	70 MPa (2 passes)	Improving the survival of probiotics with the use of alginate as a coating.	[74]
Phenolic powder from strawberry pomace	-	50 and 70 MPa (3, 5, 7 passes)	High encapsulation efficiency	[76]
*L. salivarius* spp. *Salivarius*	Mandarin juice impregnated in apple	70 MPa (2 passes)	The final count of *L. salivarius* spp. *Salivarius* encapsulation was high enough to exert a potential beneficial effect.	[77]

(-) indicates that the component has not been included in a food matrix.

**Table 5 molecules-25-03305-t005:** Research works in which HPH treatment was applied to probiotic cells.

Food Matrix	Microbial Strain	Conditions	Results	References
Yogurt	*L. Delbrueckii* ssp. *bulgaricus* LB- 12, S. *Salivarius* ssp. *thermophilus* ST-M5 and *L. acidophilus* LA-K	0, 3.45, 6.90, 10.34 and 13.80 MPa	Improved tolerance to acid and bile	[78]
-	*L. acidophilus Dru* y *L. paracasei* A13	0.1 and 50 MPa	Increased probiotic characteristics in vivo; no modification in the interaction of lactobacilli with the small intestine.	[79]
-	*Lactobacillus paracasei* A13, *Lactobacillus acidophilus* 08 and Dru, *Lactobacillus delbrueckii* spp. *lactis* 200	50 MPa	Increased functional characteristics depending on the type of strain.	[12]
Fermented milks	*Lactobacillus rhamnosus* BFE5264, *L. delbrueckii* spp. *bulgaricus* FP1 and *Streptococcus thermophilus* LI3	60 MPa	Reduced product clotting time and increased viability of the probiotic strain.	[80]
Cacciotta cheese	*Lactobacillus paracasei* A13	50 MPa	Increase in quality and decrease in cheese maturation time.	[81]
Mandarin juice	*L. salivarius* spp. *Salivarius*	0, 20 and 100 MPa	Improvement of cellular hydrophobicity.	[66]
Clementine juice	*L. salivarius* spp. *Salivarius*	25, 50, 100 and 150 MPa	Improvement of the antioxidant properties of the juice.	[82]
Fresh Culture (1% *v*/*v*)	*Lactobacillus paracasei* A13	50, 150, 200 MPa	Increase in the unsaturation in membrane fatty acids.	[83]

(-) indicates that the component has not been included in a food matrix

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
