# Peer review of "High Homogenization Pressures to Improve Food Quality, Functionality and Sustainability"

_molecules, 2020, doi:10.3390/molecules25143305_

Round 1

Reviewer 1 Report

This MS review HPH (high pressure homogenization) on the food quality and functionality. However,

  1. The reference 3 cited in this MS, seems majorly talk about HPP(high hydrostatic pressure ) effect on the juice, not HPH (please refer to its references). In page2, the author mentioned the HPH and UHPH for the pressures from 0-400 MPa, which is quite similar to the pressure range of HPP. Since HPP has been approved by FDA as a non-thermal way of pasteurization, I suggest the author clarify the difference between HPH and HPP , so the reader will not get confused.
  2. There are quite a few review papers related to HPH. What is the contribution of this MS?
  3. Table 1-5 cannot be found in the text. The number of references is quite confusing. They should be rearranged.

Author Response

THANKS FOR HELPING US TO IMPROVE THE QUALITY OF OUR MANUSCRIPT WITH YOUR REVISION. THE RESPONSES TO EACH COMMENT ARE DETAILED POINT BY POINT BELOW.

This MS review HPH (high pressure homogenization) on the food quality and functionality. However,

  1. The reference 3 cited in this MS, seems majorly talk about HPP(high hydrostatic pressure ) effect on the juice, not HPH (please refer to its references). OK, IT IS TRUE. I HAVE CHECKED THE REFERENCE IS NOT NECESSARY AND IT HAS BEEN DELETED

  1. In page2, the author mentioned the HPH and UHPH for the pressures from 0-400 MPa, which is quite similar to the pressure range of HPP. Since HPP has been approved by FDA as a non-thermal way of pasteurization, I suggest the author clarify the difference between HPH and HPP , so the reader will not get confused. OK, THE DIFFERENCE HAS BEEN CLARIFIED IN LINES 37-39.

  1. There are quite a few review papers related to HPH. What is the contribution of this MS? THE MAIN CONTRIBUTION OF THE MANUSCRIPT IS TO SHOW THE DIRECTION OF FOOD RESEARCH IN HPH IN THE LAST DECADE AND TO ARRANGE PUBLISHED ARTICLES ACCORDING TO THE MAIN CHALLENGES IN FOOD AREA. THE OBJECTIVE DESCRIBED IN THE MANUSCRIPT HAS BEEN COMPLETED TO CLARIY THE CONTRIBUTION (LINES 68-69): The objective of this work is to review how the need to combine safety, functionality and sustainability has conditioned the applications of high pressure homogenization technology in food. ADVANCES AND MAIN APPLICATION IN LAST DECADE HAVE BEEN ARRANGED ATTENDING TO THE MAIN CHALLENGES IN FOOD AREA.”

  1. Table 1-5 cannot be found in the text. The number of references is quite confusing. They should be rearranged. OK, CITATION TO TABLES HAVE BEEN INCLUDED IN THE TEXT AND REFERENCES CORRECTLY ARRANGED.

Reviewer 2 Report

There are comments on the word file provided.

Please, be careful with references. Scientific names are not in italics in many of them.

Review all the manuscript, several references must be in the format of author [reference number]. Right now you just have the number.

Author Response

THANKS FOR HELPING US TO IMPROVE THE QUALITY OF OUR MANUSCRIPT WITH YOUR REVISION. THE RESPONSES TO EACH COMMENT ARE DETAILED POINT BY POINT BELOW.

This paper is a review on the use of high pressure homogenization to achieve several goals in food. The topic is actual. The review is well done. There are some issues concerning the text.

First of all, there are five tables in the manuscript, but none of them are cited in the text. All the tables must be cited in the text. OK, CITATION TO TABLES HAVE BEEN INCLUDED IN THE TEXT.

The format of the tables makes them confusing to read. OK, THE INTERMEDIATE EDGES OF THE TABLES HAVE BEEN INCLUDED TO FACILITATE THEIR READING

Line 86, be more specific about the 74-87 % increase, to what increase are authors referring?

I would like that graphs of figure 1 to have a name in the axes. Explain more Figure 1, only after analyzing and making calculations, and I realize how the table was calculated. I suggest that the title of Figure 1 must contain a better description. OK, IT HAS BEEN INCLUDED IN THE TEXT (LINES 87-91) HOW “% INCREASE” IS DEFINED AND CALCULATED. THE NAME IN “Y” AXE HAS BEEN INCLUDED. THE PERIODS OF REPRESENTED BARS ARE SPECIFIED IN “X” AXE.

The authors must review the format of the tables. Cell content is vertically centered; then, it is difficult to read the column results; it is difficult to see where the information starts of ends. I suggest that the content of the cells should be aligned at the top of the cell. OK, THE INTERMEDIATE EDGES OF THE TABLES HAVE BEEN INCLUDED TO FACILITATE THEIR READING.

Line 145-146 S. cerevisiae must be in itàlics. OK, IT HAS BEEN CORRECTED

Line 147 review this sentence “In apricot juice a significant viability decreases (2.2 Log10 cfu/ml) was obtained” it is not understandable OK, IT HAS BEEN CORRECTED

Line 153 Zygosaccharomyces bailii must be in italics  OK, IT HAS BEEN CORRECTED

Line 154 Nevertheless, Author….. [16]. Format of reference as author (reference number) OK, IT HAS BEEN CORRECTED

Line 161 destruction[21,22]. A space is missing between the word and the bracket. OK, IT HAS BEEN CORRECTED

Lines 163-164 review this sentence “an inlet temperature of 55 °C of inlet temperature was enough” OK, IT HAS BEEN CORRECTED

Line 181 a space is missing “interaction[24].” OK, IT HAS BEEN CORRECTED

Line 188 Pseudomonas fluorescens must be itàlics OK, IT HAS BEEN CORRECTED

Line 194 L. monocitogenes must be italics OK, IT HAS BEEN CORRECTED

Line 199 an space is missing “subjectedto” OK, IT HAS BEEN CORRECTED

Line 220, who is “They”? OK, IT HAS BEEN CORRECTED

Line 233 delete “concentration” OK, IT HAS BEEN DELETED

Line 235 “90%.[41]” space is missing, and the format of the reference must be author [reference number] OK, IT HAS BEEN CORRECTED

Line 236 Flammulina velutipes must be in italics OK, IT HAS BEEN CORRECTED

Line 244 “ones.[45]” space is missing, and the format of the reference must be author [reference number] OK, IT HAS BEEN CORRECTED

Lines 244 245 review the sentence “that for the preparation of soluble peach fibre from fresh peach marc it must be dispersed” I THINK THE SENTENCE IS CORRECT.

Line 263 reference format author [63] OK, IT HAS BEEN CORRECTED

Line 280, review the phrase, “They observed the pressure required was lower the lower the fat percentage and they”. Insert the reference again, because it is not clear to whom you refer with “they” OK, IT HAS BEEN CORRECTED

Line 359 L. acidophilus must be in italics OK, IT HAS BEEN CORRECTED

Line 457 Zygosaccharomyces bailii in italics OK, IT HAS BEEN CORRECTED

Line 406 F. velutipes in italics OK, IT HAS BEEN CORRECTED

Line 502 Rosa canina in italics OK, IT HAS BEEN CORRECTED

Line 518 Ipomoea batatas in italics OK, IT HAS BEEN CORRECTED

Line 545 Brassica oleracea in italics OK, IT HAS BEEN CORRECTED

Line 548 Arenga pinnata in italics OK, IT HAS BEEN CORRECTED

Line 590 L. salivarius spp. Salivarius italics OK, IT HAS BEEN CORRECTED

Line 598 L. salivarius spp. Salivarius in italics OK, IT HAS BEEN CORRECTED

Line 600 Lactobacillus paracasei italics OK, IT HAS BEEN CORRECTED

Line 604 Lactobacillus acidophilus in italics OK, IT HAS BEEN CORRECTED

Line 619 Lactobacillus paracasei in itàlics OK, IT HAS BEEN CORRECTED

Round 2

Reviewer 1 Report

I think it is a good paper after revised now.

Reviewer 2 Report

many references still do not have scientific names in italics. This is the list

Reference 27

Reference 32

Reference 36

Reference 41

Reference 43

Reference 48

Reference 50

Reference 56

Reference 59

Reference 74

Reference 77

Reference 78

Reference 79

Please, check all of them